# The Influence of Angle of Attack on the Icing Distribution Characteristics of DU97 Blade Airfoil Surface for Wind Turbines

Chuanxi Wang [1], Weirong Lin [1], Xuefeng Lin [1], Tong Wu [1], Zhe Meng [2], Anmin Cai [1], Zhi Xu [3,*], Yan Li [3] and Fang Feng [4,*]

1   China Huaneng Clean Energy Research Institute, Beijing 102209, China; cx_wang@qny.chng.com.cn (C.W.); wr_lin@qny.chng.com.cn (W.L.)
2   Huaneng Renewables Corporation Limited, Beijing 100036, China
3   College of Engineering, Northeast Agricultural University, No. 600, Changjiang Road, Xiangfang District, Harbin 150030, China; liyanneau@neau.edu.cn
4   College of Arts and Sciences, Northeast Agricultural University, No. 600, Changjiang Road, Xiangfang District, Harbin 150030, China
*   Correspondence: zxu@neau.edu.cn (Z.X.); fengfang@neau.edu.cn (F.F.); Tel.: +86-451-55191140 (Z.X.)

**Abstract:** This study explores the influence of angle of attack (AOA) on the icing distribution characteristics of asymmetric blade airfoil (DU97) surfaces for wind turbines under icing conditions by numerical simulation. The findings demonstrate a consistence between the simulated ice shapes and experimental data. The ice thickness distribution on the lower surface of the leading edge exhibits a trend of first rising and then declining along the chord direction while showing a gradually decreasing trend on the upper surface. The ice distribution range on the upper surface of the trailing edge is broader than that on the lower surface. The peak ice thickness at the trailing edge rises significantly as AOA increases from 5° to 10°, and at the leading edge raises dramatically at droplet sizes of 30–40 μm and wind speeds of 5–10 m/s. The peak ice thickness is more significantly influenced by AOA than by ambient temperature due to the combined effect of airflow characteristics induced by AOA and latent heat (phase change) and sensible heat (thermal convection and thermal radiation) caused by ambient temperature. The findings offer valuable insights into the flow and heat transfer physics, and can operate as references for wind turbine anti/de-icing technology.

**Keywords:** wind turbine; DU97 blade airfoil; angle of attack; icing distribution characteristics; numerical simulation





## 1. Introduction

After years of rapid development, wind energy has become one of the most mature and commercially promising forms of renewable energy [1]. Wind power generation has become the main utilization method for wind energy worldwide. To improve power generation, wind turbines are installed in cold regions with high altitude due to high air density [2]. Nevertheless, owing to frequent occurrence of extreme climate conditions, wind turbine blades can experience an icing phenomenon that results in reduced power production, reduced service life, disrupted blade aerodynamics and safety hazards induced by ice shedding [3,4]. Therefore, investigating icing distribution characteristics on blade surfaces under various icing conditions is a cutting-edge topic of common concern to researchers in the field of wind energy [5], which contributes to the development of anti/de-icing design technology for wind turbine blades.

The icing wind tunnel experiment is a common approach used to explore the icing distribution characteristics. Li et al. [6] reported the icing characteristics on the NACA0018 blade airfoil at different angles of attack; Shu et al. [7] obtained actual ice distribution on the blade under atmospheric icing conditions; Gao et al. [8,9] described the influence of ice distribution on aerodynamic characteristics of wind turbine blades at different angles of

attack; Jin and Virk [10] reported that the ice shapes on the surface of the S832 airfoil were more complex than those on the surface of the S826 airfoil under the wet icing condition; and Hu et al. [11] proposed a novel method for measuring ice accretion thickness on the wind turbine blade.

However, it proved challenging to examine the ice distribution of full-size models due to limited icing wind tunnel size. Thus, it was crucial to obtain the ice distribution of the scaled models that corresponded to full-size models. Owing to indistinct flow and heat transfer physics, it was also difficult to obtain similarity parameters in experiments.

During the last few decades, numerical methods, which can provide better understanding of flow and heat transfer physics regarding ice accretion, have received widespread attention. For instance, Homola et al. [12] reported that ice accumulation on the surface of the blade airfoil significantly degraded aerodynamic characteristics under the condition of higher angles of attack; Jin and Virk [13] compared the icing characteristics of NACA0012 and NACA23012 airfoils at various angles of attack; Wang et al. [14] analyzed the aerodynamic performance of iced airfoils at different angles of attack; Hann et al. [15] reported the effect of angle of attack on the aerodynamic performance of iced S826 airfoil; Baizhuma et al. [16] obtained the icing distribution characteristics of the NACA0015 and NACA0018 airfoils at different azimuthal angles; and Ibrahim et al. [17,18] predicted the quantity of ice accumulation on wind turbine blade surfaces at various angles of attack.

According to the literature above, limitations in the size of icing wind tunnel experiments mean that only scaled models can be selected to examine the effect of icing characteristics distribution on the blade surface, and it is therefore very important to obtain the similarity relationship between scaling and full-size models. This requires a deeper understanding of the mechanism of heat and mass transfer. The numerical simulation provides a highly effective method to obtain more physical parameters, which contributes to better understanding of icing characteristics. Most researchers have analyzed the effect of AOA on the icing characteristics of different blade airfoil surfaces, but the above study has certain limitations, which follow:

(1) The effect of AOA on the icing distribution characteristics of the DU97 blade airfoil under icing conditions has not yet been identified.

(2) There are few reports on the mechanism of heat transfer and flow for iced blade airfoils that has been affected by AOA.

The aims of this study follow: (a) to use numerical simulation to explore the influence of AOA on the icing distribution characteristics of DU97 blade airfoil under icing conditions; and (b) to reveal the mechanism of heat transfer and flow for iced blade airfoils.

## 2. Experiment

### 2.1. Setup

The reflux icing wind tunnel was built at Northeast Agricultural University, and encompassed a spray system, refrigeration system, air supply system and a test section with cross-section dimensions of 250 mm × 250 mm, as illustrated in Figure 1. The DU97 blade airfoil with a chord length of 0.1 m and that was made of glass fiber reinforced plastics, was selected in the experiment, as shown in Figure 2. The span and maximum thickness of the DU97 airfoil were 20 mm and 30 mm, respectively, the range of achievable flow velocities in the icing wind tunnel was 1–15 m/s, and the blockage ratio at AOA = 12 degree was 0.67%. The turbulence intensity at 5 m/s, 10 m/s and 15 m/s were 3.8%, 3.5% and 3.3%, respectively.

Before conducting the experiment, the DU97 blade airfoil, after being cleaned with alcohol and dried with a blower, was fixed on the holder of the test section. The water droplets sprayed from the spray system mixed with cold air passing through the refrigeration system, forming supercooled water droplets that impinged on the surfaces of the blade airfoil in the test section. The temperature and achievable flow velocity were regulated through the control panel connected to the refrigeration system and air supply system, respectively. The liquid water content (LWC), with droplet median volume diameter

(MVD), was regulated through the control panel connected to the spray system. The icing conditions employed in the experiment included a flow velocity of 10 m/s, an ambient temperature of 268 K, a LWC of 3.2 g/m³ (with MVD of 26 μm) and an icing time of 300 s. The angle of attack was 12° in the experiment.

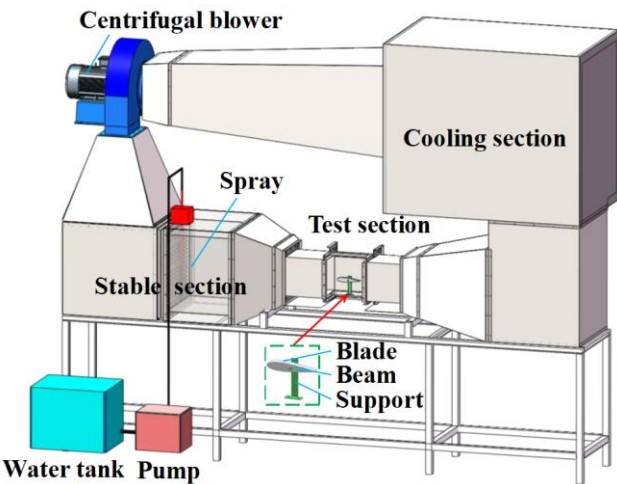

**Figure 1.** Schematics of the experimental setup.

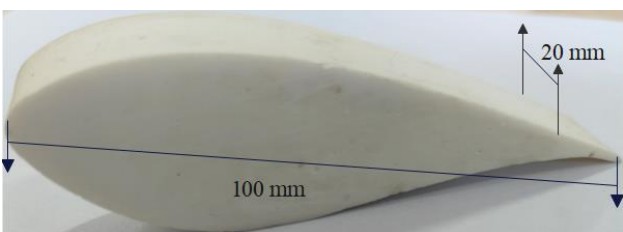

**Figure 2.** The DU97 blade airfoil.

*2.2. Experimental Procedure*

The blade airfoil installed in the test section was pre-cooled for 30 s to ensure that the surface temperature of the blade airfoil was the same as the test temperature. When the icing conditions reached the set value, the spray system installed in the icing wind tunnel began to work. The supercooled water droplets from the spray system then, under the effect of inertial force, impinged on the leading edge. When the icing time reached 300 s, the spray system stopped working. The transient ice shape on the surface of the blade airfoil was obtained by a high-speed camera (Phantomv5.1): the frame rate was 1200 frames per second, which captured the transient ice accretion shape. The ice accretion contour was then identified based on the images imported into CAXA CAD 2018 software.

## 3. Modeling

*3.1. Mathematical Model*

The airflow characteristics around the blade airfoil were obtained by the finite volume method. The water droplets flow characteristics impinging on a blade airfoil could be obtained by a Eulerian two-fluid method that took into account droplet volume fraction [19–23]. When water droplets impinged on a blade airfoil, the water film and ice accretion appeared at the leading edge. To explore the icing characteristics of blade airfoil surface, a model was developed that encompassed the water film flow and mass and energy conservation equations.

The water film flow with surface roughness was built, and water film velocity was simplified as a linear distribution normal for roughness wall, which was written as follows [24–28]:

$$\vec{V_{\text{w}}} = \frac{y\rho_{\text{a}}V_{\infty}^{2}}{2\mu_{w}}[3.476 - 0.707\ln(k_{\text{s}}/s)]^{-2.46} \tag{1}$$

$$k_{\text{s}} = 0.0012(0.43 + 0.0044V_{\infty})(0.05T_{\text{s}} - 11.27)\left[0.57 + 0.25(LWC) + 1.26(LWC)^{2}\right] \tag{2}$$

where $\vec{V_{\text{w}}}$ was the water film velocity, $\rho_{\text{a}}$ was the air density, $V_{\infty}$ was the free flow velocity, $\mu_{w}$ was the dynamic viscosity, $y$ was the distance normal to wall, $k_{\text{s}}$ was the surface roughness, $T_{\text{s}}$ was the surface temperature, and $LWC$ was the liquid water content.

The mass conservation equation included the mass transfer of the impinging water droplets, water evaporation, and ice accretion, and was written as follows:

$$\rho_{\text{w}}\left[\frac{\partial f}{\partial t} + \nabla \cdot \left(\vec{V_{\text{w}}}f\right)\right] = \alpha\left(\vec{V_{\text{d}}} \cdot \vec{n}\right)(LWC) - \frac{0.622h_{c}}{c_{a}}\left(\frac{p_{v,w} - p_{v,e}}{p_{e} - p_{v,w}}\right) - \dot{m}_{\text{ice}} \tag{3}$$

where $\rho_{\text{w}}$ was the water film density, $f$ was the water film thickness, $\alpha$ was the volume fraction of droplets, $V_{\text{d}}$ was the droplet velocity, $h_{c}$ was the convective heat transfer coefficient, $c_{a}$ was the air specific heat capacity, $\dot{m}_{\text{ice}}$ was the ice accumulation rate per unit area, and $p_{e}$, $p_{v,w}$ and $p_{v,e}$ were boundary layer edge pressure, saturated vapor pressure of water film surface and boundary layer edge, respectively.

The energy conservation equation included the heat transfer induced by the impinging water droplets, water evaporation, ice accumulation, radiative heat transfer, convective heat transfer and anti-icing heat fluxes, which was written as follows:

$$\rho_{\text{w}}\left[\frac{\partial f c_{w}\widetilde{T}}{\partial t} + \nabla \cdot \left(\vec{V_{\text{w}}}f c_{w}\widetilde{T}\right)\right] = \alpha\left(e_{d} + \frac{1}{2}\vec{V_{\text{d}}}^{2}\right)\left(\vec{V_{\text{d}}} \cdot \vec{n}\right)(LWC)$$
$$- \frac{0.622h_{c}L_{\text{evap}}}{c_{a}}\left(\frac{p_{v,w} - p_{v,e}}{p_{e} - p_{v,w}}\right) + \dot{m}_{\text{ice}}\left[L_{\text{fus}} - c_{\text{ice}}\left(\widetilde{T}_{\text{ice}} - \widetilde{T}_{0}\right)\right] \tag{4}$$
$$+ \sigma\varepsilon\left(T_{\infty}^{4} - T_{\text{w}}^{4}\right) + h_{c}\left[\gamma\frac{V_{\infty}^{2}}{2c_{a}} - (T_{\text{s}} - T_{\infty})\right] + \dot{Q}_{\text{anti}}$$

where $c_{w}$ was the water film specific heat capacity, $\widetilde{T}$ was the interface equilibrium temperature, $e_{d}$ was the droplet energy, $L_{\text{evap}}$ was the evaporation of latent heat, $L_{\text{fus}}$ was the fusion of latent heat, $c_{\text{ice}}$ was the ice-specific heat capacity, $\sigma$ was the Stefan-Boltzmann constant, $\varepsilon$ was the emissivity, $\dot{Q}_{\text{anti}}$ was anti-icing heat flux, and $T_{\text{s}}$, $T_{\text{w}}$ and $T_{\infty}$ were the surface temperature, water film temperature and free flow temperature, respectively.

Owing to the fact that the water film thickness, quantity of ice accumulation and interface equilibrium temperature were unknown, additional equations were built to solve unknown variables in conjunction with the above equations, which could be described as follows:

$$\begin{cases} f \geq 0 \\ \dot{m}_{\text{ice}} \geq 0 \\ f\widetilde{T} \geq 0 \\ \dot{m}_{\text{ice}}\widetilde{T} \leq 0 \end{cases} \tag{5}$$

For Equation (4), the recovery factor was solved based on flow regime, including $n = 1/2$ for laminar regime and $n = 1/3$ for turbulent regime, which was written as follows:

$$\gamma = 1 - \left(\frac{V_{k}}{V_{\infty}}\right)^{2}(1 - \text{Pr}^{\text{n}}) \tag{6}$$

where $\gamma$ was the recovery factor, and $V_k$ was the wind speed related to roughness.

The heat transfer coefficient was solved based on laminar and turbulent regions. For the laminar region, the heat transfer coefficient was written as follows [29]:

$$h_c = 0.296 \frac{k_a}{\sqrt{\nu}} \left[ V_e^{-2.88} \int_0^s V_e^{1.88} ds \right]^{-1/2} \tag{7}$$

where $k_a$ was air thermal conductivity, $V_e$ was velocity of boundary layer edge, and $\nu$ was the kinematic viscosity.

To determine laminar-turbulent transition, the Von Doenhoff criterion was employed as follows [30]:

$$\text{Re}_k = \frac{V_k k_s}{\nu_1} \geq 600 \tag{8}$$

When the Reynolds number exceeded 600, the boundary layer was considered to be turbulent flow [29], which was written as follows:

$$\frac{V_k}{V_e} = \frac{2k_s}{\delta} - 2\left(\frac{k_s}{\delta}\right)^3 + \left(\frac{k_s}{\delta}\right)^4 + \frac{1}{6}\frac{\delta^2}{\nu}\frac{dV_e}{ds}\frac{k_s}{\delta}\left(1 - \frac{k_s}{\delta}\right)^3 \tag{9}$$

where $\delta$ was the boundary layer thickness, which was given by [29]:

$$\delta = \frac{315}{37V_e^3}\sqrt{0.45\nu \int_0^s V_e^5 ds} \tag{10}$$

The relationship between heat transfer coefficient and the Stanton number was written as follows:

$$h_c = \text{St}\rho C_p V_e \tag{11}$$

The empirical relationship for solving the Stanton number was written by [31]:

$$\text{St} = 0.5C_f / \left( \text{Pr}_t + \sqrt{0.5C_f}\text{St}_k^{-1} \right) \tag{12}$$

$$\text{St}_k = 1.92\text{Re}_k^{-0.45}\text{Pr}_t^{-0.8} \tag{13}$$

### 3.2. Geometry and Computational Mesh

The computational domain of DU97 blade airfoil is illustrated in Figure 3. To obtain the external flow field of the blade airfoil in a large space, the size of the fluid region was set to approximately 20 times the chord length, before the near wall enhancement approach with inflation factor (1.1) was performed to obtain non-uniform structured mesh on DU97 blade airfoil. It was observed that the computational average relative error between 91,475 and 194,827 mesh was less than 0.5%, as depicted in Figure 4. The computational efficiency for 91,475 mesh improved by 52%, compared to 194,827 mesh. The mesh of 91,475 was therefore employed to ensure computational accuracy while also improving computational efficiency in numerical simulation. On the basis of the book [32], the operating conditions of wind turbines included an ambient temperature of 233–323 K, a wind speed of 3–25 m/s, and LWC of 0–5 g/m$^3$ (with MVD of 10–5000 µm). In this study, the icing conditions included wind speed of 5–15 m/s, ambient temperature of 258–268 K, LWC of 0.6 g/m$^3$ (with MVD of 30–50 µm) and an icing time of 1800 s.

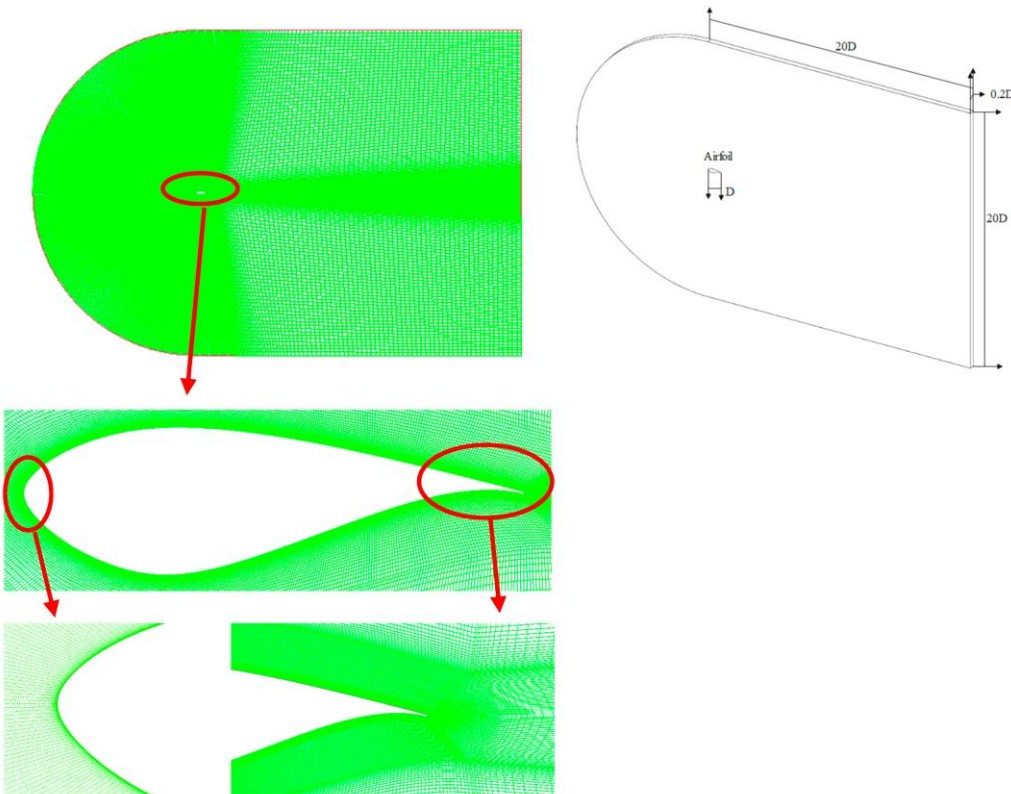

**Figure 3.** Computational domain and meshing of DU97 blade airfoil. Red circles represent the zoomed view of the mesh at leading and trailing edges of the airfoil.

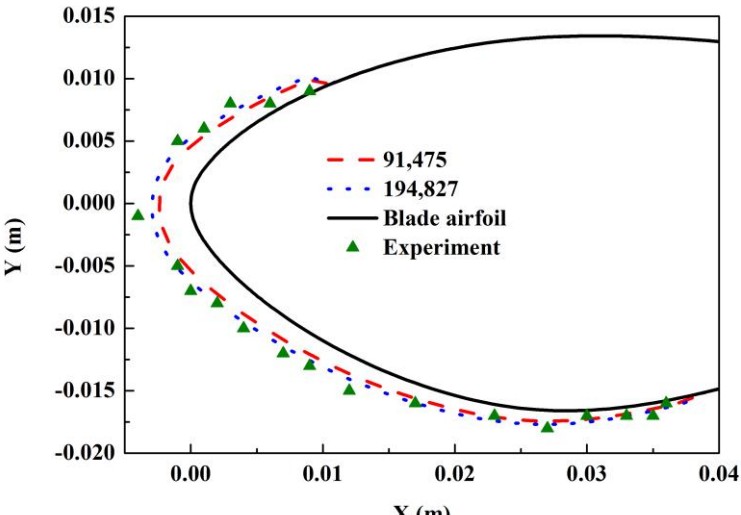

**Figure 4.** Mesh independence.

*3.3. Boundary Condition and Solution Method*

The far-field condition was set as the boundary condition of fluid field around the DU97 blade airfoil, and the surface of the DU97 blade airfoil was set as a boundary condition of no slip wall in a numerical simulation (the detailed boundary conditions are illustrated in Table 1). The heat and transfer model was, after considering roughness effect, implemented by compiling UDF, which coupled with FENSAP-ICE to achieve the numerical calculation of icing characteristics. The detailed discrete format and iterative solution method described by [5] was then applied.

**Table 1.** Boundary conditions.

| Type | Values |
|---|---|
| Far-field | Pressure: 101,325 Pa<br>Temperature: 258–268 K<br>Speed: 5–15 m/s<br>LWC: 0.6 g/m$^3$ |
| Wall | No slip |

### 3.4. Model Validation

To validate the numerical method, the icing experiments on a DU97 blade airfoil were conducted in the Northeast Agricultural University icing wind tunnel. It was discovered that a consistence between the simulated ice shapes and experimental data was obtained, as shown in Figure 5. The relative error between simulation and experiment was in the range of 1.7%–8.3%. It was discovered that the error of the vast majority of points was within 2%. Significantly, the maximum error point was at the ice horn. This phenomenon aligned with the literature, as reported by Lu et al. [33]. The maximum error (8.3%) in this paper was smaller than that reported in the literature (more than 10% error) [14,33] due to the consideration of roughness effect in the water film heat and mass transfer model. The ice shape exhibited asymmetry distribution due to the interaction between geometry configuration and the angle of attack. The ice distribution on the surface of the blade airfoil fluctuated in the experiment and was relatively smooth in the numerical simulation, which was ascribed to the fact that the medium volume diameter of droplets was employed in the numerical simulation, which was equivalent to droplets of different sizes.

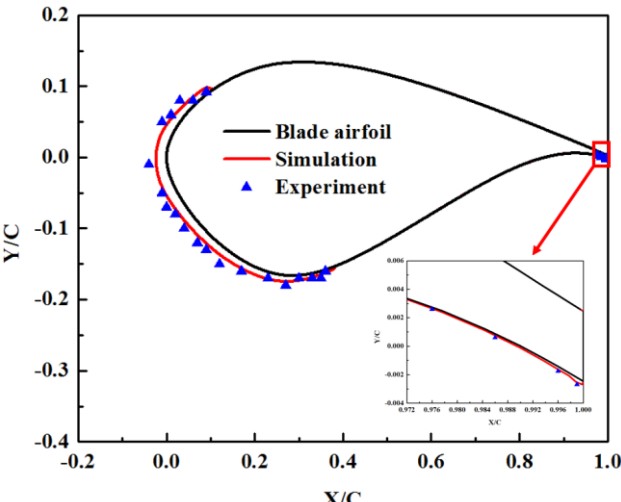

**Figure 5.** The simulated and experimental ice shape on the blade airfoil surface.

## 4. Results and Discussion

Investigating the influence of AOA on the icing distribution characteristics on the surface of DU97 blade airfoil can provide fundamental guidance for de/anti-icing technology applied during the operation of a horizontal axis wind turbine. Figure 6 depicts the ice shapes on the surface of the DU97 blade airfoil at different AOA for an ambient temperature of 258 K. It is evident that the ice shape on the surface of the blade airfoil exhibits more significant asymmetry with an increase in AOA, which results in a greater degradation of aerodynamic performance, which aligns with findings reported by Homola et al. [12]. This is ascribed to the change in airflow induced by geometry configuration, and heat flux caused by phase change, convective heat transfer and thermal radiation. In addition, it is prone to form ice horns with an increase in AOA, due to the distribution of non-uniform impinging water droplets on the blade airfoil surface.

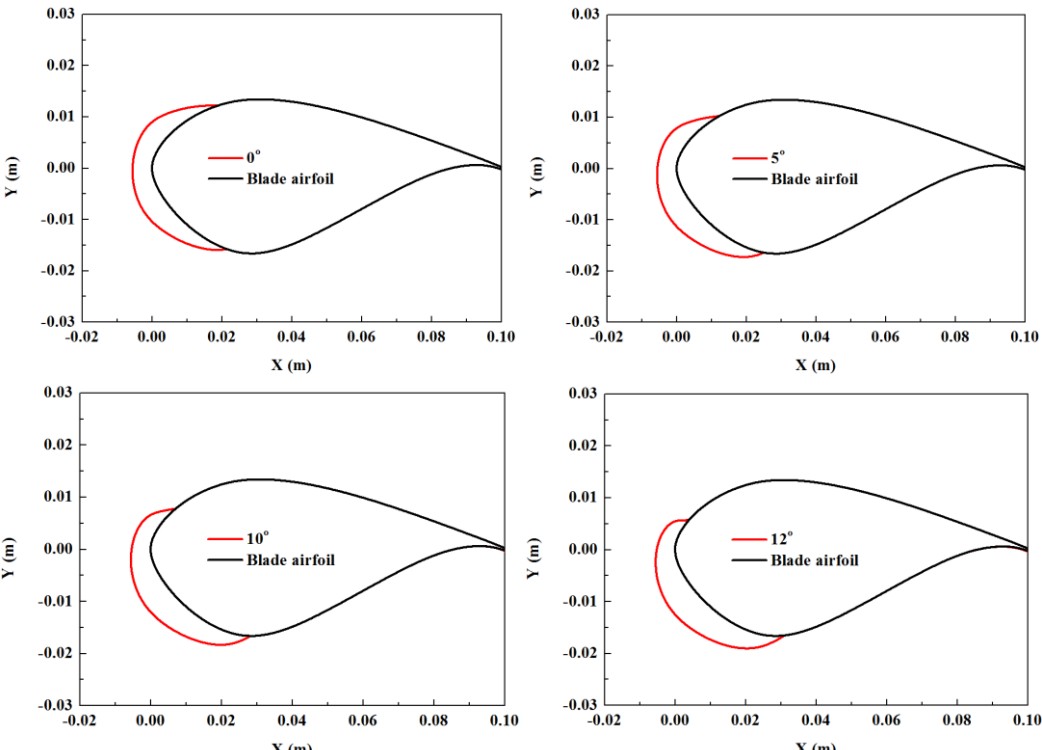

**Figure 6.** The ice shapes at various angles of attack.

Figure 7 illustrates the ice thickness distribution on the upper and lower surfaces of DU97 blade airfoil along the X-direction at AOA, and shows that the peak ice thickness rises from 5.99 mm to 6.93 mm with an increase in AOA (0–10°). For the lower surface of the leading edge, the distribution of ice thickness exhibits a trend of first rising and then declining along the X-direction. On the other hand, the distribution of ice thickness in the upper surface of the leading edge shows a gradually decreasing trend. In addition, it is evident that a small amount of ice accumulation appears on the trailing edge of the DU97 blade airfoil as AOA increases, which is significantly different from NACA0012 blade airfoil [5]. Notably, the peak ice thickness on the trailing edge of the DU97 blade airfoil rises significantly as AOA increases from 5° to 10°, which is due to the fact that the droplet collection efficiency on the surface of the blade airfoil, which is induced by the interaction between geometry configuration and angle of attack, changes, leading to the change in heat and mass transfer caused by phase change, convective heat transfer and thermal radiation.

Figure 8 shows the ice thickness distribution on the surface of the DU97 blade airfoil along the Y-direction at angles of attack of 0–10°, and indicates that the peak ice thickness on the upper surface of the blade airfoil occurs at an AOA of 0° and at an AOA of 10° on the lower surface. In addition, the ice thickness on the lower surface of the trailing edge is significantly higher than that on the upper surface. Notably, the ice accretion distribution range on the upper surface of the trailing edge is broader than that on the lower surface, which is due to the fact that the change in AOA causes the variation in airflow characteristics, resulting in non-uniform droplet collection distribution on the blade airfoil surface. Subsequently, under the combined effects of phase change, convective heat transfer and thermal radiation, the ice distribution on the blade airfoil surface undergoes corresponding changes.

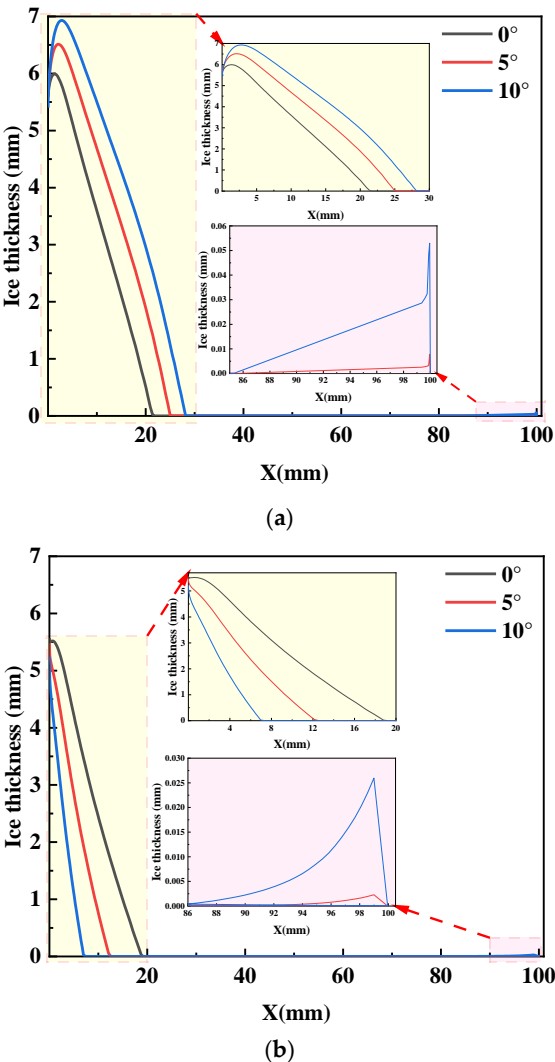

(**a**)

(**b**)

**Figure 7.** The ice thickness distribution along the X-direction at AOA, (**a**) The lower surface; (**b**) The upper surface.

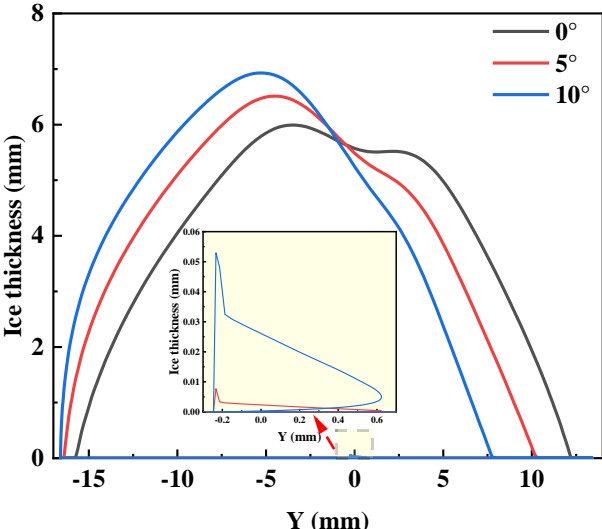

**Figure 8.** The ice thickness distribution along the Y direction at various angles of attack.

Figure 9 shows ice thickness distribution on the surface of DU97 blade airfoil along the X-direction for droplet sizes of 30–50 μm, and indicates that the peak ice thickness rises from 5.8 mm to 6.9 mm as droplet sizes increase from 30 μm to 50 μm. Owing to an increase in droplet sizes, the droplet collection efficiency on the surface of DU97 blade airfoil raises, resulting in an increase in ice thickness through solidification. It is observed that the peak ice thickness increases more significantly for droplet sizes of 30–40 μm than for droplet sizes of 40–50 μm, which is because larger droplets colliding with the blade airfoil will break and splash, causing a slow increase in the peak droplet collection efficiency on the surface of the DU97 blade airfoil. In addition, it is evident that ice thickness distribution exhibits a trend of first rising and then declining at the leading edge and trailing edge. This is due to the fact that the droplet collection efficiency of the blade airfoil caused by the interaction between geometry configuration and droplet size varies, leading to the change in ice thickness distribution through solidification.

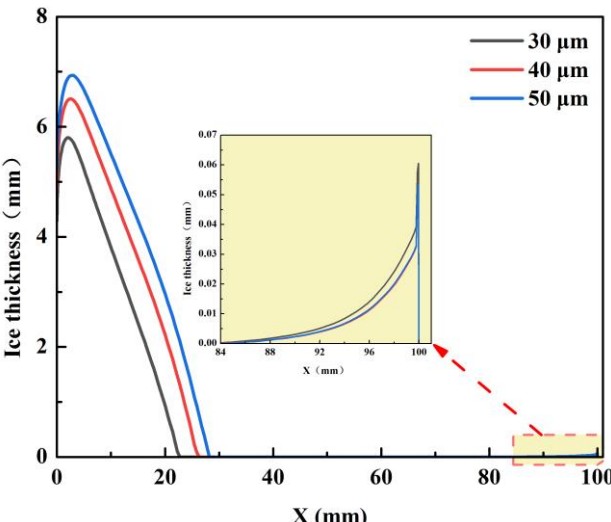

**Figure 9.** The ice thickness distribution along the X-direction at various droplet sizes.

Figure 10 depicts ice thickness distribution on the surface of DU97 blade airfoil along the X-direction at wind speeds of 5–15 m/s, and shows that the peak ice thickness rises from 3.6 mm to 9.8 mm as wind speed rises from 5 m/s to 15 m/s. Owing to an increase in wind speed, the droplet collection efficiency of the blade airfoil surface caused by inertial force raises, resulting in increased ice accumulation through solidification. It is evident that the peak ice thickness increases more significantly at wind speeds of 5–10 m/s than at wind speeds of 10–15 m/s. This is elucidated by the fact that higher inertial force causes droplet deformation, leading to a change in ice distribution caused by the droplet collection efficiency. Additionally, it is observed that ice thickness distribution exhibits a trend of first rising and then declining, at both the leading edge and trailing edge. This is because the droplet collection efficiency of the blade airfoil induced by the interaction between geometry configuration and airflow characteristics varies, leading to non-uniform ice thickness distribution through solidification.

The effects of ambient temperature and AOA on the peak ice accretion thickness are explored, as Figure 11 shows. The peak ice accretion thickness varies in the range of 5.72–6.93 mm in relation to ambient temperature (−15−−5 °C) and angle of attack (0–10°). As ambient temperature decreases from −5 °C to −15 °C, the peak ice accretion thickness increases by 4.7%, 7.4% and 12.1% at angle of attack of 0°, 5° and 10°, respectively. Additionally, when an angle of attack increases from 0° to 10°, the peak ice accretion thickness raises by 8.1%, 13.7% and 15.7% at ambient temperature of −5 °C, −10 °C and −15 °C, respectively. Notably, the peak ice accretion thickness is more significantly influenced by angle of attack than by ambient temperature, which may be attributed to the following fac-

tors: (1) ambient temperature affects the freezing rate of water droplets on the blade airfoil surface; and (2) AOA results in non-uniform droplet collection distribution on the blade airfoil surface, which is induced by the variation in airflow characteristics. Furthermore, ice accretion thickness on the blade airfoil surface exhibits a nonlinear distribution, which is due to the combined effects of latent heat (phase change) and sensible heat (thermal convection and thermal radiation).

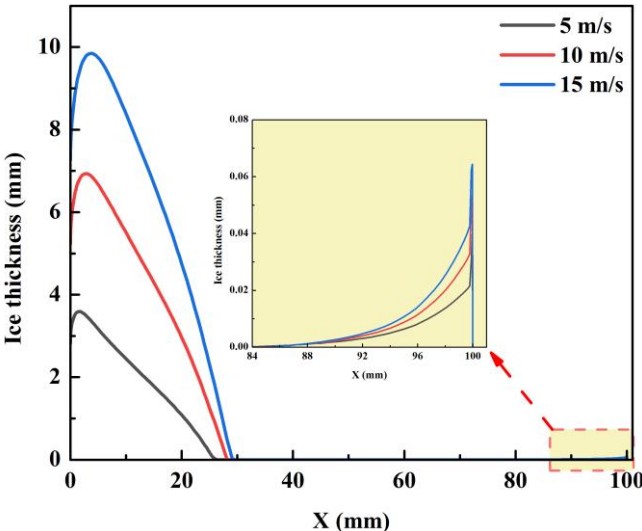

**Figure 10.** The ice thickness distribution along the X-direction at various wind speeds.

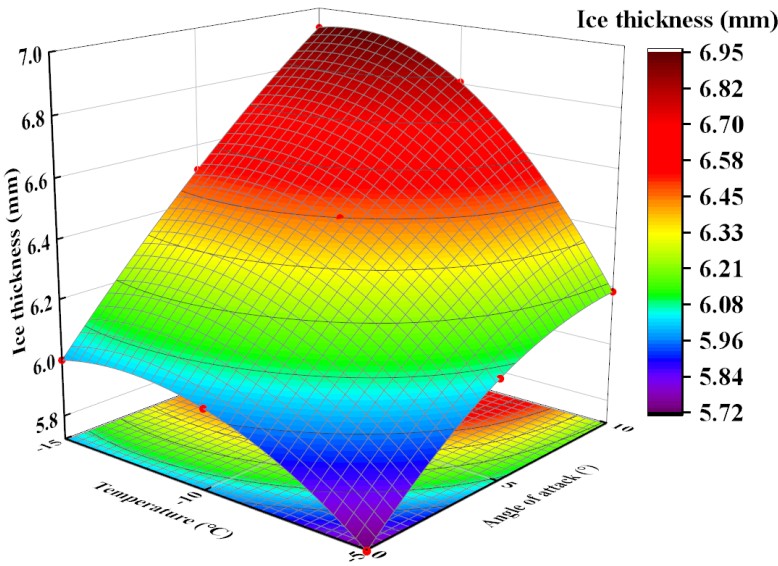

**Figure 11.** Peak ice thickness affected by ambient temperature and angle of attack.

## 5. Conclusions

In this study, numerical simulation was used to explore the influence of angle of attack (AOA) on the icing distribution characteristics of asymmetric blade airfoil (DU97) surfaces under icing conditions, and the analysis of the mechanism of heat transfer and flow for iced blade airfoils yielded the following findings:

(1) The ice thickness distribution on the lower surface of the leading edge exhibits a trend of first rising and then declining along the chord direction, and the counterpart on the upper surface shows a gradually decreasing trend.

   (2)    The peak ice accretion thickness on the trailing edge of DU97 blade airfoil rises significantly as AOA increases from 5° to 10°.

   (3)    The ice distribution range on the upper surface of the trailing edge is broader than on the lower surface, which is due to the non-uniform droplet collection distribution on the blade airfoil surface.

   (4)    The peak ice thickness raises more dramatically for droplet sizes of 30–40 μm than droplet sizes of 40–50 μm.

   (5)    The peak ice thickness increases more significantly at wind speeds of 5–10 m/s than at wind speeds of 10–15 m/s.

   (6)    The peak ice accretion thickness is more significantly influenced by angle of attack than by ambient temperature, which is due to the combined effects of airflow characteristics induced by angle of attack and latent heat (phase change) and sensible heat (thermal convection and thermal radiation) induced by ambient temperature.

The present study mainly focuses on the effect of icing distribution characteristics on a static DU97 blade airfoil and provides insight into macroscopic flow and heat transfer physics. In fact, the size of the wind turbine blade is very large, and the icing characteristics on the blade surface will become more complex under rotating conditions. It is very difficult to investigate the ice distribution on a full-size wind turbine blade, and a deeper understanding of the icing physical mechanism is therefore crucial for proposing reasonable similarity criteria. In future, a multi-scale icing model with the rotating framework will be developed to reveal ice distribution from the perspectives of multi-scale flow and heat transfer physics, which will provide an important point of reference for the exploration of reasonable similarity criteria.

**Author Contributions:** Conceptualization, C.W. and Z.X.; methodology, F.F.; software, Z.X. and F.F.; validation, W.L. and T.W.; formal analysis, X.L.; data curation, Z.M.; writing—original draft preparation, C.W.; writing—review and editing, Z.X. and F.F.; visualization, A.C.; supervision, W.L.; funding acquisition, Y.L. and Z.X. All authors have read and agreed to the published version of the manuscript.

**Funding:** This research was funded by the Research and Application of Key Technologies for High-Efficient Anti-icing and De-icing of Wind Turbine Blades of China Huaneng Group (HNKJ22-H101) and the National Natural Science Foundation of China (Grant No. 52106228).

**Institutional Review Board Statement:** Not applicable.

**Informed Consent Statement:** Not applicable.

**Data Availability Statement:** Data are contained within the article.

**Conflicts of Interest:** Authors Chuanxi Wang, Weirong Lin, Xuefeng Lin, Tong Wu and Anmin Cai were employed by the company China Huaneng Clean Energy Research Institute. Author Zhe Meng was employed by the company Huaneng Renewables Corporation Limited. The remaining authors declare that the research was conducted in the absence of any commercial or financial relationships that could be construed as a potential conflict of interest.

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
