# Peer review of "The Influence of Angle of Attack on the Icing Distribution Characteristics of DU97 Blade Airfoil Surface for Wind Turbines"

_coatings, doi:10.3390/coatings14020160_

Round 1

Reviewer 1 Report

Comments and Suggestions for Authors

1. The experimental results are only used for validation purposes, and not included in the results and discussion section thus it should not be claimed in the abstract that this study is an experimental study. If so, do explain them in Results and discussion section. On the other hand only three simulation cases of AOA 0, 5 and 10 degree are discussed in results section. Only three cases are not enough to considered the articles for publication in a journal. it is suggested to vary the size of the droplets and flow velocities in the simulation/experiments and analyze the results. With only three cases it can be a conference paper and cannot be considered as journal article. 

2. what was span and maximum thickness of the foil?

3. Provide more details about the tunnel, e.g. turbulence intensity etc. at various flow velocities

4. What is the maximum and minimum range of achievable flow velocities in the wind tunnel? provide the blockage ratio at AoA =12 degree.

5. Please mention the scale on the diagram, or complete dimensions of the foil

6. What frame rate was used to capture the transient ice accretion shape?

7. Is it a 2-D computation or 3-D? why the results in simulation were not compared with experimental results for mesh independence study? Please provide a comparison of the computational results with mesh 91475 and 194827 to that of the experimental. 

8. What is difference in the foil in fig. 2 and in fig. 3? if its the same only one figure is enough.

9. Please provide the dimensions of the flow domain in dimensionless form

10. Please provide the zoomed view of the mesh at leading and trailing edges of the foil as well.

11. No need to rewrite/repeat the same details in model validation section 3.4 as those in section 2.1 and 2.2.

12. Please provide the pointwise %error in the results and plot the same to see the accuracy of the simulated results. this is very important as far as the acceptance or rejection of the article is concerned. Please do provide some of the published data where results differ in similar %age for evaluation.

13. What is the AoA for the results compared in this figure?

14. it is provided that the experiments and simulations both were carried out at AOA=12 degree but the same is not provided in Fig. 6

15.  Results and discussion is merely around 1500 -2000 words, too less of a content. more results as mentioned in point 1 can be added to improve/expand this section.

Reviewer 2 Report

Comments and Suggestions for Authors

The present manuscript studies the influence of angle of attack on the icing distribution char- acteristics of asymmetric blade airfoil surfaces for wind turbines under icing conditions by numerical simulation and icing wind tunnel experiment. The present work seems to be very interesting in the recent trend of research and applications in wind energy sectors. However, the present version needs a revision before considering for publication. My comments and suggestions are as follows:

1. A robust motivation paragraph should be added at the beginning of the Introduction to justify the present work in the specific area of research, aim, objective, and application.

2. Introduction: Last paragraph should be discussed a bit more.

3.  The Figure 2 missing of labelling.

4. The variables are used in Eqs. (1-13) are not explained in the texts.

5. The computational cost of the numerical results should be discussed.

6. In Figure 5, the percentage of agreement between the model results need to be discussed to access the level of accuracy.

7. Conclusion: The limitations and the future scope of the present study need to be discussed.

Comments on the Quality of English Language

Minor editing of English language is required.

Round 2

Reviewer 2 Report

Comments and Suggestions for Authors

No more further comments.

Comments on the Quality of English Language

Still, minor editing of the English language is needed.